# Characterization of Lactic Acid Bacteria Strains Isolated from Algerian Honeybee and Honey and Exploration of Their Potential Probiotic and Functional Features for Human Use

**DOI:** 10.3390/foods12122312

**Published:** 2023-06-08

**Authors:** Meriem Meradji, Nadia Bachtarzi, Diego Mora, Karima Kharroub

**Affiliations:** 1Laboratoire de Recherche Biotechnologie et Qualité des Aliments (BIOQUAL), Institut de la Nutrition, de l’Alimentation et des Technologies Agro-Alimentaires (INATAA), Université Frères Mentouri Constantine 1 (UFMC1), Route de Ain El Bey, Constantine 25000, Algeria; meriem.meradji@umc.edu.dz (M.M.);; 2Department of Food Environmental and Nutritional Sciences (DeFENS), University of Milan, 20122 Milan, Italy

**Keywords:** *Apis mellifera* intermissa, honey, *Apilactobacillus* sp., *Fructobacillus* sp., *Lactobacillus* sp., probiotic

## Abstract

Using culture enrichment methods, 100 strains of bacilli of lactic acid bacteria (LAB) were isolated from honeybee *Apis mellifera intermissa* and fresh honey, collected from apiaries located in the north-east of Algeria. Amongst all of the isolated LAB, 19 selected strains were closely affiliated to four species—*Fructobacillus fructosus* (10), *Apilactobacillus kunkeei* (5), *Lactobacillus kimbladii* and/or *Lactobacillus kullabergensis* (4)—using phylogenetic and phenotypic approaches. The in vitro probiotic characteristics (simulated gastrointestinal fluids tolerance, autoaggregation and hydrophobicity abilities, antimicrobial activity and cholesterol reduction) and safety properties (hemolytic activity, antibiotic resistance and absence of biogenic amines) were evaluated. The results indicated that some strains showed promising potential probiotic properties. In addition, neither hemolytic activity nor biogenic amines were produced. The carbohydrate fermentation test (API 50 CHL) revealed that the strains could efficiently use a broad range of carbohydrates; additionally, four strains belonging to *Apilactobacillus kunkeei* and *Fructobacillus fructosus* were found to be exopolysaccharides (EPS) producers. This study demonstrates the honeybee *Apis mellifera intermissa* and one of her products as a reservoir for novel LAB with potential probiotic features, suggesting suitability for promoting host health.

## 1. Introduction

Dairy-based probiotic foods are the most common probiotic products available in the global market. However, more attention has been attributed to non-dairy probiotic products of plant origin such as fruits, vegetables and cereals because of lifestyle trends (e.g., vegetarianism and veganism) and the rise in milk allergy and lactose intolerance, which is estimated at 60% of the world’s population [1]. Most commercial probiotic strains are of human gastrointestinal tract or dairy origin [2]. These latter might not be adapted to technological and physiological challenges due to the presence plant-derived substrates (cellulose, pectin, phytate, etc.). Consequently, probiotic cultures used in plant-based fermented products should preferably be isolated from plant material [3]. Furthermore, strains from intestinal and dairy sources have been found to be unable to overcome some technical problems for industrial application, especially viability rate, since commensal human bacteria are relatively more sensitive to oxygen and low pH when used in fermented non-dairy products [4]. Thus, alternative sources known as “unconventional sources” have been progressively screened for potential probiotics. 

Fruits, vegetables, flowers and traditional fermented beverages are valuable sources of functional and probiotic cultures with high potential for use in food processing to enhance the development of vegan probiotic products with improved nutritional quality and more consistent health benefits [5,6]. As part of the bacterial flora within insects, LAB are also found in flowers and insect environment.

In this regard, in the last few years, symbiotic lactic acid bacteria (LAB) microbiota were first located in the honey crop of *Apis mellifera*, and later in bee hive environments [7,8,9]. The majority of these symbionts were described as novel species [10,11]. Besides its role in the honeybee’s food production and preservation, these co-evolved microbiota have shown a protective action against severe bee pathogens involving different mechanisms such as biofilms formation, preventing gut dysbiosis, release of antimicrobial compounds and immunomodulatory properties [12,13]; in addition, they can increase the longevity of the host via rescued metabolic stress markers [14]. Another function of some bee symbionts is the alteration of the levels of biogenic amines such as serotonin and dopamine, which impact host neurophysiology and behavior [15]. Honey bee LAB are recurrently isolated from bee hive products which are a part of human diet and traditionally used to promote healing [7]. This fact, and the above-listed beneficial functions, lead us to discuss whether these bee symbionts could have the same benefits for human as for their original host. It was only in 2015 that the role of heat-killed strains of *Apilactobacillus. kunkeei* was explored in human immune modulation [16], followed by further studies reporting antimicrobial action of the 13 symbiotic bee species against pathogens causing human wound infections [17,18].

Recently, *Lactobacillus kunkeei* and *Lactobacillus apinorum*, two of nine honeybee lactobacilli, were reclassified in new genus “*Apilactobacillus*” [19]. Both species belongs to fructofilic lactic acid bacteria (FLAB), a specific subgroup of LAB [20], which tend to expand continuously [10,21]. Essentially, the FLAB group is represented only by *Lactobacillus* and *Fructobacillus* genera. To date, *Fructobacillus fructosus* is the only species of the genus *Fructobacillus* isolated from the gastrointestinal tract (GIT) of honeybees [22,23].

In fact, FLAB inhabit only fructose-rich niches and prefer fructose as their growth substrate [24,25]. This ability deserves more interest, because it could help in the avoidance of fructose-mediated irritable bowel syndrome (IBS) [26]. In addition, these bacteria are able to reduce the levels of fermentable oligosaccharides, disaccharides, monosaccharides and polyols (FODMAP_S_) in fruits, vegetables and cereals [27]. Otherwise, certain intracellular or extracellular polymeric substances produced by many species of FLAB are seen to have benefits on human health, such as EPS, which has a cholesterol-lowering ability [26]; galacto-oligosaccharides (GOS_S_), which find application as prebiotics [28]; and nicotinamide mononucleotide (NMN) for its pharmacological and anti-aging efficacies [29].

In Algeria, few studies of the associated honeybee and hive products microflora have been conducted, from which none of the symbiotic *Lactobacillus* sp. or *Fructobacillus fructosus* were reported [30,31,32,33]. 

Based on these facts, the aims of this work are to contribute to a greater knowledge of honeybee LAB microbiota isolated from the gut of Algerian honeybees (*A. mellifera intermissa)* and their honey, and also to evaluate some of their functional and probiotic properties for future human use, since none of these honeybee symbionts lactic acid bacteria are yet used in commercial human probiotic formulations.

## 2. Materials and Methods

### 2.1. Honeybees and Honey Sampling

Five bees belonging to *A. mellifera intermissa* species were obtained in sterile bottles and two fresh honey samples were taken using a micropipette and sterile microtubes from unsealed cells within a hive from apiaries located in the north-east of Algeria (36°21′0″ N 6°35′60″ E) and from colonies maintained using standard beekeeping practices, during 2019. Both honey samples were analyzed for moisture, acidity and total reducing sugar, in accordance with the Food and Drug Administration (FDA) standard methods [34].

### 2.2. Isolation of Acid Tolerant LAB and Characterization of Fermentation Pattern

The whole intestinal tract (esophagus to rectum) or only nectar-filled stomachs were acquired from the bees with aseptic excision under luminal flow. The anatomical samples were placed in 0.9 mL physiological saline solution (0.9% *w*/*v* NaCl, 0.1% *v*/*v*, Tween 80, and 0.1% *w*/*v* peptone), while 1 g of each honey sample was suspended in 9 mL of the saline solution. A ten-fold serial dilution was performed, and pure cultures were obtained on modified MRS (de Man, Rogosa, and Sharpe) agar medium (Oxoid, UK) supplemented with 2% (*w*/*v*) fructose and 0.1% (*w*/*v*) L-cysteine. The isolates were cultivated anaerobically using anaerobic jars with Anaerocult A gas packs (Merck, Darmstadt, Germany) at 37 °C for 72 h [8]. 

Using a basic chemical test for an initial screening of Lactobacillus [35,36], only Gram-positive, catalase-negative and rod-shaped isolates were kept and maintained in modified MRS broth supplemented with 30% (*v*/*v*) glycerol then stored at −40 °C.

According to Viernstein et al. [37], a primary screening for acid tolerance was performed in modified MRS broth adjusted to pH 2.5 with 1N HCl for 90 min at 37 °C. The determination of survival rate was carried out via single streaking on modified MRS agar plates. Isolates that grew on the agar after 24–48 h under anaerobic incubation at 37 °C were considered to be acid tolerant strains. 

The isolates retained were assessed for their carbohydrate fermentation patterns using an API 50 CHL system kit, according to the manufacturer’s instructions (bioMérieux SA, Marcy l’Etoile, France).

### 2.3. Molecular Identification and Phylogenetic Analysis

Molecular identification was performed as described by Mora et al. [38]. An aliquot of 400 µL of a fresh modified MRS broth, culture corresponding to approximately10^9^ cells, was subjected to alkaline lysis and DNA extraction. DNA quantity and quality evaluation were carried out using a spectrophotometer (NanoDrop ND-1000, Thermo Fisher Scientific, Waltham, MA, USA). The 16S rRNA gene was amplified via polymerase chain reaction (PCR) using universal bacterial primers: P0-Forward (5′-GAAGAGTTTGATCCTGGCTCAG-3′) and P6-Reverse (5′-CTACGGCTACCTTGTTACGA-3′). PCR products were electrophoresed in Tris-acetate-EDTA buffer and photographed in UV light, then purified using a QIAquick PCR purification kit (Qiagen, Hilden, Germany) according to the manufacturer’s instructions. Purified DNA was sequenced by Microsynth Seqlab Co. (Maschmühlenweg Göttingen, Germany) using primer 1100R.

The resulting 16S rRNA gene sequences of identified strains were investigated using the nBLAST tool and the EzBioCloud Database version 2018-05 [39]. Nucleotide sequences were deposited in the NCBI database under accession numbers from OP848250 to OP848268.

### 2.4. Assessment of Proteolytic, Lipolytic Activities and EPS Production

Proteolytic activity was tested by spotting 2 µL of LAB suspensions onto skim milk agar medium and incubating at the optimal growth temperature for 4 days; a clear zone around each colony was an indication of this phenomenon [40]. Lipolytic activity was indicated by the presence of a limpid halo around the colonies grown on buffered modified MRS medium (pH 7–7.4) supplemented with 1, 2 and 3% (*w*/*v*) of Tween-80 [41]. Concerning the production of EPS, it was carried out in MRS agar supplemented with 15% (*w*/*v*) sucrose. Ropy or mucoide large colonies were detected if EPS was produced [42].

### 2.5. Resistance to Simulated Gastro-Intestinal Conditions

Tolerance to gastric intestinal conditions was carried out according to [43,44]. Strains were grown in modified MRS overnight at 37 °C, then centrifuged (5000× *g* for 15 min at 4 °C), and bacterial pellets were washed twice with phosphate-buffered solution (PBS). Afterwards, bacterial cells were re-suspended in PBS at a final concentration of 10^8^ CFU/mL. A volume of 0.2 mL of the bacterial suspension was added to 1.8 mL of simulated gastric solution at pH 2 and 3 (NaCl 6.2 g/L, KCl 2.2 g/L, CaCl_2_ 0.22 g/L, NaHCO_3_ 1.2 g/L, 0.3% (*w*/*v*) pepsin) and another aliquot of 0.2 mL of bacterial suspension was added to 1.8 mL of simulated intestinal solution at pH 7.5 (NaCl 6.2 g/L, KCl 2.2 g/L, CaCl_2_ 0.22 g/L, NaHCO_3_ 1.2 g/L, 0.1% (*w*/*v*) pancreatin, 0.3% (*w*/*v*) bile salts). Cells in 0.1 M PBS pH 6.5 were used as control. After 2 and 4 h of incubation in gastric and intestinal juices, respectively, viability testing was performed via plate counts on modified MRS agar after 48 h at 37 °C under anaerobic conditions.

### 2.6. Autoaggregation and Cell Surface Hydrophobicity

Autoaggregation was determined according to the method described by Cozzolino et al. [45]. An overnight culture of each strain was centrifuged (5000× *g* for 15 min at 4 °C), after which the cells were washed three times with PBS and re-suspended in sterile PBS to obtain 10^8^ CFU/mL. Samples were retained to stand for 24 h at room temperature. Then, the OD 600 nm (Optizen POP, Mecacys Co., Ltd., Seongnam, Republic of Korea) of the upper layer was measured and expressed as autoaggregation (%) = [(OD0 − OD24)/OD0] × 100. Bacterial cell surface hydrophobicity was assessed through cell adhesion to xylene (Merck) [46]. Briefly, the strain suspended in sterile PBS (10^8^ CFU/mL) was mixed with xylene (3/1, *v*/*v*) and stirred for 2 min. The absorbance of the aqueous phase (A1) was measured at 600 nm after one hour at room temperature (24 °C ± 1). Hydrophobicity was expressed as [1 − (A1/A0)] × 100. A0 and A1 are the absorbance values of the aqueous phase before and after addition of xylene.

### 2.7. Antimicrobial Activity

The accuracy of antimicrobial activity testing was monitored via parallel use of five human pathogenic referenced strains: *Staphylococcus aureus* ATCC 43300, *Escherichia coli* ATCC 25522, *Pseudomonas aeruginosa* ATCC 27853, *Klebsiella pneumonia* ATCC 700603 and *Bacillus cereus* ATCC 10876.

First, these pathogens were used as target bacteria and tested for their resistance to common antibiotics used in medical practice using the disc diffusion method (as shown in the antibiotic susceptibility section). Then, antimicrobial activity of the LAB strains against the above-mentioned pathogens was evaluated using the agar well diffusion assay for broth culture (BC) and cell-free supernatant (CFS), according to Testa et al. [47]. In short, 20 mL of both MRS and Mueller–Hinton soft agar (0.7% (*w*/*v*) agar) were inoculated with 200 µL (10^8^ CFU/mL) of each overnight culture of pathogens. A quantity of 50 µL of an overnight broth culture (BC) of each strain was placed into a 5.0 mm drilled well. In parallel, cell-free supernatant neutralized with 1N NaOH to pH 6.5 and catalase treated for one hour at 0.1 mg/mL (2000–5000 U/mg protein, Sigma, Steinheim, Germany) were recovered by centrifugation at 5000× *g* for 15 min and sterilization with a 0.22 μm filter (VWR Co. Europe, Darmstadt, Germany). A quantity of 50 µL of each treaded CFS was placed into a well. Sterile modified MRS was used as negative control. All plates were kept at 4 °C for 2 h prior to incubation. After 24 h of incubation at 37 °C, the zone of inhibition (ZOI) was measured [48].

### 2.8. In Vitro Cholesterol Lowering Ability

Determination of cholesterol lowering ability was conducted using a colorimetric method [49]. In brief, MRS broth containing 0.3% bile salts and 0.01% cholesterol (100 µg/mL) was inoculated with 1% (*v*/*v*) of each strain (10^8^ UFC/mL). Non-inoculated broth was used as control. After 24 h of incubation at 37 °C, 1 mL of 33% KOH (*v*/*v*) and 3 mL of ethanol were added to 1 mL of culture supernatant and the contents were mixed after the addition of each component. The tubes were heated for 15 min at 60 °C. After cooling, 2 mL of distilled water and 3 mL of hexane were added and vortexed. Tubes were allowed to stand for 20 min at room temperature. A volume of 1 mL of hexane layer was transferred to a fresh tube and allowed to dry completely; residues were resuspended in 1.5 mL ferric chloride reagent (Sigma-Aldrich, Steinheim, Germany). After 10 min, 1 mL of concentrated sulfuric acid (Sigma-Aldrich, Steinheim, Germany) was added. The mixture was vortexed and placed in the dark for 45 min. The absorbance was measured at 560 nm. A standard curve using cholesterol was used to determine cholesterol concentration and the percentage of cholesterol lowering was calculated using the following formula:

Cholesterol lowering % = [(cholesterol in control medium) − (cholesterol in test medium)] × 100/(cholesterol in control medium)

### 2.9. Antibiotic Susceptibility

A total of 13 antibiotics, used in therapy of common infections, were tested through the disc diffusion method [50]. A concentration of 10^8^ CFU/mL pure cultures of the LAB strains was swabbed on modified MRS agar plates. Then, the antibiotic discs (Liofilchem Sri, Roseto degli Abruzzi, Italy) were placed on the surface of the plates. Finally, the diameter of the zone of inhibition around the discs was measured after 24 h of incubation at 37 °C. Results were interpreted according to Clinical and Laboratory Standards Institute [51].

### 2.10. Hemolytic Activity

The selected LAB isolates were cultured on blood agar supplemented with 5% (*v*/*v*) sheep blood at 37 °C for 48 h. The plates were analyzed for the presence of green-hued (α-hemolysis), white or transparent zones (β-hemolysis) or the absence of hemolysis (γ-hemolysis) around the colonies [46].

### 2.11. Biogenic Amines Production

To detect bacteria producing biogenic amines, a qualitative test was used [43]. First, strains were inoculated at 1% (*v*/*v*) in modified MRS broth containing 0.1% (*w*/*v*) of each amino acid precursor (L-lysine, tyrosine disodium salt, L-histidine monohydrochloride and L-ornithine monohydrochloride (Sigma Aldrich, Darmstadt, Germany)) and incubated at 37 °C for 18 h. The screening was carried out using a specially formulated agar medium (0.5% tryptone, 0.5% yeast extract, 0.5% meat extract, 0.25% NaCl, 0.05% glucose, 0.1% Tween 80, 0.02% MgSO_4_, 0.005% MnSO_4_, 0.004% FeSO_4_, 0.2% ammonium citrate, 0.001% thiamine, 0.2% K_2_PO_4_, 0.01% CaCO_3_, 0.005% pyridoxal-5-phosphate, 1% amino acid precursor, 0.006% bromocresol purple, 2% agar; pH 5.3). For this, 10 µL of each culture was spotted on the medium containing the same amino acids precursor. After 4 days of incubation at 37 °C, a positive result was indicated by the color change and by amino acid precipitation around the corresponding spot for tyramine only. The assays were conducted in duplicate.

### 2.12. Statistical Analysis 

Results were presented as the mean ± standard deviation of three values. Statistical differences were analyzed using one-way ANOVA followed by multiple mean comparisons Tukey’s test. A *p* value <0.05 was considered statistically significant. Statistical analysis was performed using GraphPad Prism 9.4.1(681) software.

## 3. Results and Discussion

### 3.1. Physico-Chemical Characterization of Honey Samples 

The first honey sample used in the present study included 18.4% moisture, 80% total reducing sugar and pH 4.31, while 18.8% moisture, 79.6% total reducing sugar and pH 4.72 were recorded for the second honey sample. Regarding potential hydrogen, our values are consistent with international standards limit (pH 3.40–6.10) and with those previously reported for other honey samples from Algeria [34,52,53,54]. The acidity of honey is related to the amount of ionisable acid therein and its mineral composition. This acidity contributes to the flavor and the stability of honey against microbial spoilage. The determination of water content is an indication of the state of maturation of honey. In general, the moisture content should not exceed 20% for harvested honey. According to Gonnet [55], only honeys with a water content of less than 18% are good to keep. The two samples in this study showed moisture contents greater than 18%. These high values can be explained by the early harvest of honey, i.e., before its complete ripening, which was the objective of sampling in this study. The moisture of some studied Algerian honey was from 14.4 to 17.8% [53,54]. Unlike stored honey, fresh honey represents a naturally fermented food with a great diversity and concentration of viable LAB species; this may explain honey’s benefits when consumed directly after being harvested throughout history. 

### 3.2. Molecular Identification and Phylogenetic Analysis

A total of 100 bacilli were isolated from two honey samples and five honeybees (three digestive tracts and two full crops). From these, 19 strains showed a better growth on modified MRS agar at pH 2.5. The acid-resistant isolates were subjected to phylogenetic analysis, of which 9 originated from the digestive tract of honeybees and 10 from honey samples (Figure 1). The 16S rRNA gene sequences were compared to the EzBioCloud and Genbank. The results showed that the isolated LAB belonged to the *Lactobacillaceae* family [19]. Sequence identities ranged from 99.04 to 100% between the selected strains and type strains. Isolates were affiliated with 4 species: *Fructobacillus fructosus* (10), *Apilactobacillus kunkeei* (5), *Lactobacillus kimbladii* and *Lactobacillus kullabergensis* (4). These are a part of the honeybee microbiota, which is dominated by nine to ten phylotypes [8,22,56,57], within the predominant genera *Lactobacillus* (formerly) and *Bifidobacterium* [58,59]. These microbiota play a key role in the production of honey [7,22] and bee bread [60,61,62], which is in consistent with our findings since two identified species (*Apilactobacillus kunkeei* and *Fructobacillus fructosus*) were found in both honeybees and their honey. This was explained by the addition of bacteria during the process through which nectar becomes honey [8]. Two phylotypes identified from the nine generally abundant in honeybee microbiota belong to the genera *Lactobacillus* and *Apilactobacillus* of the phylum *Firmicutes*, named “Firm-4” and “Firm-5”. Several members of these two phylotypes have been described as novel species [63]. The same findings were reported for different or identical phylotypes of the genus *Lactobacillus* originating from honeybees guts and their food [58,64,65]. Recently, three and two species belonging to *Apilacobacillus* and *Lactobacillus*, respectively, were described as novel and hosting the *Apis mellifera* gut [10,66,67]. *Lactobacillus kimbladii* and *Lactobacillus kullabergensis*, belonging to the Firm-5 clade, were commonly identified as the most numerically abundant phylotype in honey bee guts [68,69]. As these two species are 98.9% identical at their 16S rRNA locus, ML10, AB5, AB18 and AB46 were found to belong to one of these two species [70]. Additionally, in a recent study [66], results based on 16S rRNA gene sequence analysis, phylogenetic tree based on concatenated sequences and phylogenomic tree based on whole genome sequences indicated that three strains isolated from the gut of honeybee (*Apis mellifera*) were most closely related to *L. kullabergensis* and *L. kimbladii*, having 99.1 to 99.7% 16S rRNA gene sequence similarities, whereas nucleotide identity (ANI), digital DNA–DNA hybridization (dDDH) and average amino acid identity (AAI) values could differentiate the three studied strains to two novel species as *Lactobacillus huangpiensis* sp. nov. and *Lactobacillus laiwuensis* sp. nov. Additional genetic-based methods would be ideal in deciding whether to confirm the taxonomic position of ML10, AB5, AB18 and AB46 strains or to reclassify them as new species. *Fructobacillus fructosus*, as the only *Fructobacillus* species isolated from GIT of honeybees, and *Apilactobacillus kunkeei* are representative of FLAB, a specific subgroup of LAB [71,72]. They live in symbiosis with insects that have a fructose-rich diets, such as honeybees, and also in hive products [21]. To the best of our knowledgem this is the first report of the presence of these four species in Algerian honeybee and fresh honey [30,33]. 

### 3.3. Biochemical Characterization

With regard to carbohydrates fermentation pattern (Table 1), all the strains within the four species were able to completely ferment 17 to 20 out of 49 tested substrates. AB5 was the only strain capable of fermenting arabinose. In our study, FLAB showed composite fermentation profiles, as also reported by other authors [44,62,73]. Moreover, certain pentose described as toxic to honeybees were found to be metabolized by our tested strains [74,75]. However, several other studies reported that FLAB metabolize only a limited number of carbohydrates [23]; hence, carbohydrates metabolized in the API 50 CHL test are usually two to four. In fact, FLAB were able to metabolize the three common nectar sugars (fructose, glucose and sucrose). Acid production from D-galactose, mannitol, melibiose, trehalose and potassium gluconate are strain dependent [71]. Pentoses, arabinose, ribose, and xylose, which are usually metabolized by heterofermentative LAB, were not utilized by FLAB, probably due to the absence of epimerases or isomerases [76]. As seen from the carbohydrate utilization profile, each of the four lactobacilli strains showed a different spectrum of metabolic capacities among some tested sugars: tagatose was fermented by AB46 and AB18, while arabinose and mannitol were fermented only by AB5. According to Ellegaard et al. [70], this divergence among strains of the same species can be explained by the presence of 100–250 protein families per strain overall, which were assigned to the carbohydrate metabolism and transport (COG) category, of which about 60 were conserved among all strains of the fifth phylotype. Notably, in the same study, fructose and glucose were the sole carbon sources that promoted growth of *L. kimbladii* and *L. kullabergensis* strains. Additionally, *L. kimbladii* possessed the ability to ferment mannose.

### 3.4. Assessment of Proteolytic, Lipolytic Activities and EPS Production

Proteolytic activity was indicated by the presence of a clear zone around colonies in skim milk agar medium. All tested strains exhibited moderate proteolytic activity. Regarding lipolytic activity, all strains affiliated to *A. kunkeei* species exhibited this feature, while only 2 of the 10 *F. fructosus* were positive, namely ML17 and AM. Accordingly, protease or lipase production was detected in *A. kunkeei* and *F. fructosus* [77,78], whereas Filannimo et al. [62] did not observe protease activity in *F. fructosus*. We could therefore consider the proteolytic and lipolytic activities a strain-specific feature in *A. kunkeei* and *F. fructosus* strains. To improve the nutritional value of food products by increasing the bioactive compounds content (phenolic compounds, short chain fatty acids, bioactives peptides, etc.), the production of related enzymes is a positive trait in probiotics to provide health benefits to hosts [79].

Strains U21 and AB1, affiliated to *A. kunkeei*, and strains ML17 and AM, affiliated to *F. fructosus*, were the only EPS producers. EPS are the principal component in extracellular polymeric substances and, when secreted into the environment, provide protection to bacteria; they also play a role in cellular recognition and host colonization [80]. However, despite their technological features, it has been suggested that exopolysaccharides produced by food grade organisms (GRAS), in particular LAB, may confer health benefits in humans [73,81]. Ellegaard et al. [70] reported that no EPS clusters were identified within the Firm 5 group, similar to our findings since no EPS production was detected for the tested *L. kimbladii* and *L. kullabergensis* strains. Otherwise, since some tested strains belonging to *F. fructosus* and *A. kunkeei* were EPS producers, glucansucrases (responsible for the glucan type EPS production) were accordingly identified in some *L*. *kunkeei* strains [81,82]. Additionally, genes encoding for putative polysaccharide biosynthesis and GtfC glucosyltransferase gene were present in *A. kunkeei* genome [83,84]. Additionally, genomic analysis revealed that the levansucrase gene was detectable in *Fructobacillus* spp. genomes, but genes encoding for sucrose-specific phosphotranspherase system (PTS), sucrose phosphorylase and dextransucrase were absent [85]. Importantly, unlike the findings of Endo and his colleagues, EPS production was detected in *A. kunkeei* and *F. fructosus* [20,86].

### 3.5. Resistance to Simulated Gastro-Intestinal Conditions

Simulation of gastric conditions did not induce relevant changes in counts for all the most tested strains at pH 3 after 120 min; a viability rate from 69.70 to 99.53% was recorded, while only 14 of 19 strains survived at pH 2, presenting a viability rate between 32.69 to 82.57%. However, significant differences were highlighted after 4 h of exposure to simulated intestinal conditions. A high loss of viability was detected for AB24 and AB34 isolates and viability rates of less than 50% for AB70 and ML17, whereas a final viability ranging from 52.56 to 86.34% was registered for the rest of the tested isolates (Figure 2). Our findings revealed similar results for other FLAB strains [26,87], although lower levels of bile salts and low pH tolerance (20 to 40% and 0 to 66%, respectively) were observed for FLAB strains of *L. kunkeei* and *F. fructosus* in a previous report [81]. 

Considering that the viability and functionality of beneficial microorganisms can be negatively affected when passing through the digestive tract, most of the positive effects are expressed in the gut, where probiotics arrive in adequate numbers. Therefore, the screening of resistance to gastrointestinal stressors remains a crucial selection criterion. Our result highlighted a survival rate equal or greater to those reported by Succi et al. [88] in different lactic acid bacterial strains marketed as probiotics. Mechanisms of resistance to acidity reported for LAB may be due to EPS or ammonia production, changes in membrane composition (fatty acids) and repair of macromolecules such as DNA and proteins by chaperones or maintaining pH homeostasis [89,90,91], whereas bile resistance is commonly linked to deconjugation of bile salts, bile efflux or EPS production [92,93].

Based on these results, *A. kunkeei* AB24 and AB34, *F. fructosus* ML17 and AB70 and *L. kimbladii/kullabergensis* AB18 strains were removed, and only 14 of 19 strains were kept for further assessment.

### 3.6. Autoaggregation and Cell Surface Hydrophobicity

The surface adhesive properties of lactic acid bacteria to intestinal epithelial cells, such as hydrophobicity and aggregation, are correlated with adhesion to abiotic and biotic surfaces [46]. In this respect, the autoaggregation and cell surface hydrophobicity of the LAB strains were determined. As can be seen in Figure 3, all the LAB strains revealed different levels of autoaggregation and hydrophobicity. Strains *L. kimbladii/kullabergensis* AB46 and *F. fructosus* AB36 showed the highest and the lowest autoaggregation level, respectively (82.29% and 25.22%), while the others had autoaggregation rates from 40.70 to 77.67%. 

Regarding hydrophobicity, the highest and the lowest values were registered by *L. kimbladii/kullabergensis* ML10 and *F. fructosus* FM (83.40% and 45.80%, respectively). A high level of cell surface properties was observed for the LAB isolates; these findings were at higher or lower levels than previous results [26,46,81,94]. Additionally, significant differences (*p* < 0.05) for cell surface (autoaggregation and hydrophobicity) capacities were observed among the strains. This confirms that these properties are also strain-dependent characteristics [94,95].

Despite the mechanism of action, the nature of the extracellular matrix of probiotics influences microbial interaction with their environment. Cell surface molecules such as proteins and EPSs are involved in several non-specific and specific extracellular interactions and cell surface ligand-receptors. Variation in the composition of surface components might be responsible for differences in cell surface properties among probiotic isolates. The findings in this study support the proposal that differences observed in hydrophobicity and aggregation levels, even within the same species, are a strain-specific trait and depend on the microbial cell surface compounds. Overall, further research must be performed in order to understand and identify the different mechanisms and components involved in these phenomena.

### 3.7. Antimicrobial Activity

The inhibitory activity of tested strains against foodborne bacteria was also evaluated to better highlight their potential as probiotics. As shown in Table 2, in particular, all broth cultures (BCs) showed a ZOI from 10 to 27 mm against *K. pneumonia*, *B. cereus* and *S. aureus*. Isolates AM, 49d, AB46 and ML17 did not show any inhibitory activity towards *E. coli*, while *P. aeruginosa* was not inhibited only by the last three. With the few exceptions of *F. fructosus* ML20 against *E. coli* and *B. cereus* and of *A. kunkeei* U21 against *S. aureus*, none of the neutralized and catalase-treated CFSs were able to prevent pathogen growth. Compared to tested conventional antibiotics, only streptomycin and gentamicin expressed a ZOI from 13 to 28 mm against the five pathogens, while 6 of the 13 antibiotics were effective against one or two indicators and the rest were totally ineffective (Table 3). In fact, the antimicrobial activity of the BCs could be attributed to LAB metabolites such as hydrogen peroxide, organic acids, ethanol, carbon dioxide, or proteinaceous compounds, which are the main antimicrobial agents [96].

In a previous study, the antibacterial activities of honeybee LAB were exploited in the treatment of chronic wounds caused by a multitude of pathogens. Specifically, the biofilm produced by *A. kunkeei* was able to mollify infections by *P. aeruginosa* [17,24]. Another study [44] reported that both cultures and CFSs of *A. kunkeei* strains showed anti-*Pseudomonas* activity, while they were ineffective against *S. aureus* and *E. coli* in contrast to our results. The antimicrobial activity exerted by neutralized and catalase-treated CFSs of *A. kunkeei* U21 and *F. fructosus* ML20 against *E. coli*, *B. cereus* and *S. aureus*, respectively, suggest that this effect was most likely due to the production of proteinaceous compounds or other antimicrobial metabolites, as previously detected in studies involving *A. kunkeei* and *F. fructosus* species or other *Lactobacillus* species [22,46,48]. In line with these, [12] investigated bacteriocin synthesized during stress conditions by 13 honeybee LAB species, from which neither *L. kimbladii* nor *L. kullabergensis* were bacteriocin producers, in contrast to the *A. kunkeei* tested strain. In this case, the inhibitory molecule was a muramidase that acts as a lysozyme by degrading the cell wall peptidoglycan of both Gram-positive and Gram-negative bacteria. Interestingly, it has been shown that these muramidases may also interact with the human immune system, acting as immune-adjuvants [25]. Ref. [97] described Kunkecin A as the first bacteriocin and lantibiotic found from FLAB and the first nisin-type lantibiotic reported in the *Lactobacillaceae* family. LAB produce ribosomally synthetized bacteriocins, proteinaceous substances that exhibit bactericidal or bacteriostatic activity against several foodborne pathogens and are degraded by gut proteases, which could make from them an alternative antibiotic agent for clinical applications.

### 3.8. In Vitro Cholesterol Lowering Ability

As indicated in Figure 4, 10 of 14 tested isolates were able to significantly reduce the initial cholesterol level by more than 50% after 24 h. Strains *F. fructosus* AB36 (5.80%), *L. kimbladii*/*kullabergensis* AB46 (16.16%) and AB5 (33.08%) were significantly different from the other strains, representing the lowest cholesterol reduction ability, while the highest cholesterol removal rates were registered for *A. kunkeei* 49d (83.08%) and *L.kimbladii/kullabergensis* ML10 (77.27%). 

High cholesterol is found to be associated with cardiovascular diseases (CVD), an important cause of mortality worldwide. Various reports corroborate our findings regarding potential cholesterol-lowering bacteria. Ref. [98] reported a significant correlation between probiotic intake and relief of cardiovascular disease risk factors, including high cholesterol. Similarly, a meta-analysis of randomized controlled trials concluded that probiotics can significantly reduce serum total cholesterol [99]. Notably, few studies reported the cholesterol lowering ability of FLAB [26,44]. Assimilation by probiotics is one of several mechanisms for cholesterol lowering, viz. bile salt deconjugation ability and bile salt hydrolase (BSH) activity, co-precipitation of cholesterol with bile acids, adhesion of cholesterol to probiotic cell wall, micellar sequestration of cholesterol, conversion of cholesterol to an insoluble compound coprostanol, and short chain fatty acid-mediated cholesterol lowering [100]. In fact, such cholesterol lowering probiotics indicate biotherapeutic potential for management of raised cholesterol level, and hence for combating CVD risks. However, as also shown in our work, these property seems to be strain-dependent [101,102].

### 3.9. Antibiotic Susceptibility

Antimicrobial disc susceptibility tests were performed in accordance with CLSI (2015) [51]. Varying resistances to 8 of 13 antibiotics was reported. All the tested strains showed resistance towards vancomycin, oxacillin and nalidixic acid and were susceptible to ampicillin, chloramphenicol and clindamycin. Of the isolates, 85.71% and 64.28% were sensitive to erythromycin and cefoxitin, respectively, while the others showed intermediate level of sensitivity. On the other hand, a less clear response (resistance–intermediate-sensitive) was observed towards gentamycin, tetracycline, penicillin, streptomycin and trimethoprim (Table 4). 

In this study, the antibiotic resistances observed are different from those reported by [103]. They pointed to the presence of an acetyltransferase responsible for chloramphenicol resistance in *A. kunkeei*, while none of our *A. kunkeii* strains were found to be resistant to this antibiotic. Similarly, refs. [44,46] found several levels of antibiotic sensitivity in fructophilic LAB. Ref. [104] suggested that *F. fructosus* MCC 3996 could be suitably prescribed during antibiotic therapy using oxacillin and streptomycin, as the strain showed resistance against these two antibiotics. In accordance, all *F. fructosus* in our study were found to be resistant to oxacillin but also sensitive or intermediate to streptomycin. In a previous study, Temmerman et al. tested the resistance of a multitude of LAB isolates recovered from 55 European probiotic products using the disc diffusion method. The result showed that 79% of the isolates were found to be resistant to kanamycin, 65% to vancomycin, 26% to tetracycline, 23% to penicillin G, 16% to erythromycin and 11% to chloramphenicol [50]. These data reflect the wide heterogeneity of antibiotic resistance in fructophilic LAB. However, antibiotic sensitivity is extremely relevant for the safety assessment of probiotics. Probiotics strains should be susceptible to at least two clinically relevant antibiotics [105], and antibiotic resistance can be tolerated if it is not acquired, and therefore transferrable by horizontal gene transfer events to harmful bacteria [106]. However, according to [107,108], the high frequency of resistance to vancomycin, gentamicin, streptomycin and trimethoprim shown by the *Lactobacillus* strains does not represent a major safety concern in itself, since intrinsic resistance is estimated to present a minimal potential for horizontal spread, making the evaluation of antibiotic markers an important safety criteria to confer the ‘qualified presumption of safety’. The above considerations make the LAB characterized in this study potentially interesting as probiotics in subjects that undergo antibiotic treatment. 

### 3.10. Hemolysis

Hemolytic activity is considered as a common virulence characteristic among pathogens due to cytolysin, a bacterial toxin expressed by some LAB, which exhibits hemolytic and bactericidal activities. The absence of cytolysin coding genes is considered a benefit for the application of LAB as probiotics and also in the food industry [109]. 

The safety of the LAB strains was confirmed in vitro via negative blood hemolytic activity, supporting their safety as potential probiotics in accordance with previous observations for other LAB [46,110,111].

### 3.11. Biogenic Amines Production

The safety of the LAB strains was also checked by assessing the bacterial decarboxylation of amino acids, which generates biogenic amines in food [112]. Neither cadaverine, tyramine nor histamine were produced by all the tested isolates. For putrescine, a slight color change was detected for all. Biogenic amines (BAs) are basic nitrogenous compounds with low molecular weight, usually produced by LAB through amino acids decarboxylation or deamination [113,114]. BAs are involved in physiological roles in living organisms, but despite the fact that their excessive production or intake can induce undesired toxicological effects, a shared regulation limiting the amounts of biogenic amines in food is still lacking [115].

The main BAs found in food are histamine, tyramine, putrescine and cadaverine [116]. Notably, the European Food Safety Authority (EFSA) confirmed histamine and tyramine as the most toxic and particularly relevant for food safety [115]. It is likely that none of our strains were producers. Therefore, it is important to include non BA-producing LAB as a requirement for strains intended to be used as starters or probiotics [117].

## 4. Conclusions

In the present study, LAB isolated from fresh natural honey and honeybees’ intestinal tracts were assessed for probiotics traits. The 19 selected isolates from the preliminary screening test fulfilled several criteria to be used as probiotic microorganisms, including a broad-spectrum antibacterial activity, adhesion to hydrocarbons, autoaggregation activity and several safety aspects such as the susceptibility to antibiotics, the absence of hemolytic activity and the absence of biogenic amines production. However, all the isolates showed in vitro high heterogeneity in gastric conditions survival and in their ability in lowering cholesterol. Thus, 14 of the 19 isolates can be considered potential probiotic strains with a beneficial role in maintaining good intestinal tract health in humans and might be involved in novel strategies against the increased number of antibiotic-resistant pathogens in human infections, or even as food biopreservatives. However, in vivo and clinical trials and further in vitro studies and characterization at genomic level are mandatory to fulfil the safety evaluation and the probiotic traits of these newly characterized LAB strains.

## Figures and Tables

**Figure 1 foods-12-02312-f001:**
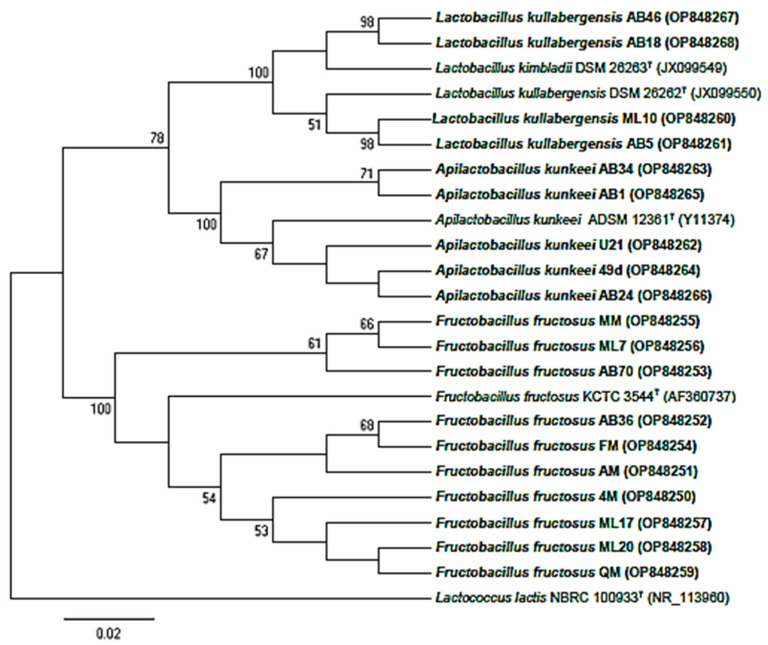
Phylogenetic tree showing the relationships among identified taxa and type strains of each species based on 16S rRNA gene sequences. *Lactococcus lactis* NBRC 100933^T^ was used as an outgroup. The tree was made using the neighbor-joining method with maximum composite likelihood model in MEGA 11. Only bootstrap values >50% are indicated. Genbank accession numbers are shown following the organism’s name in parentheses. The scale bar indicates 0.02 nucleotide changes per position. AB1, AB5, AB18, AB24, AB34, AB36, AB46 and AB70 were isolated from whole intestinal tracts; 49d was isolated from honey stomach; 4M, AM, FM, QM, MM and U21 were isolated from honey (sample 1) and ML7, ML10, ML17 and ML20 were isolated from honey (sample 2).

**Figure 2 foods-12-02312-f002:**
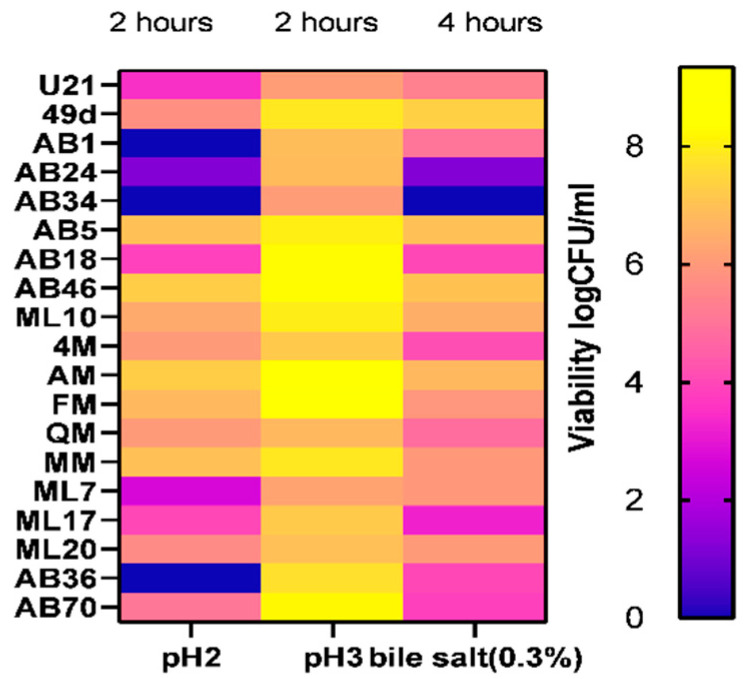
Heat map representing bacterial viability (%) of tested strains in simulated gastrointestinal conditions. Results are shown as mean ± standard deviation (*n* = 3).

**Figure 3 foods-12-02312-f003:**
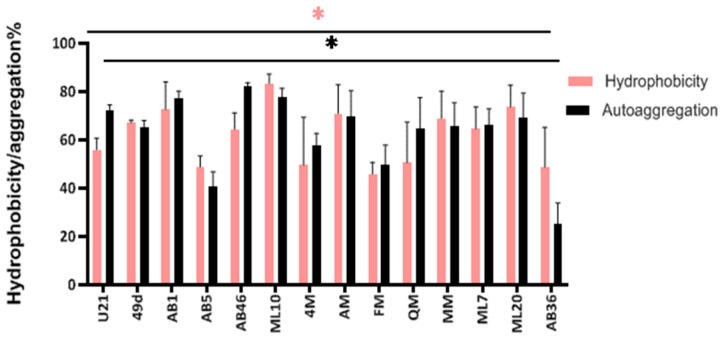
Adhesion expressed as hydrophobicity (%) and autoaggregation (%) of the tested strains measured as optical density (OD) values at 600 nm. * indicates significant differences at *p* < 0.05 despite color among hydrophobicity and autoaggregation abilities of the LAB. Results are shown as mean ± standard deviation (*n* = 3).

**Figure 4 foods-12-02312-f004:**
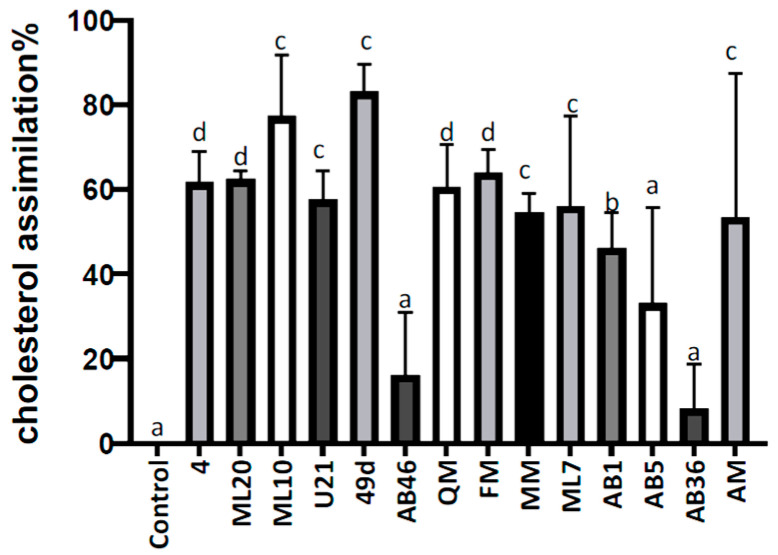
Cholesterol lowering (%) detected in cell-free supernatants from broth cultures (in MRS) inoculated with tested strains and analyzed after 24 h of incubation at 37 °C. Control, uninoculated MRS broth. Different letters indicate significant differences at *p* < 0.05 among the cholesterol lowering of the LAB. Results are shown as mean ± standard deviation (*n* = 3).

**Table 1 foods-12-02312-t001:** Carbohydrates fermented by tested strains detected through API 50 CH kit.

	U21	AB1	49d	AB24	AB34	ML10	AB5	AB18	AB46	4M	AM	FM	QM	MM	ML7	ML17	ML20	AB36	AB70
L-arabinose	-	-	-	-	-	-	+	-	-	-	w	-	-	-	-	-	-	-	-
D-ribose	+	+	+	+	+	+	+	+	+	+	+	+	+	+	+	+	+	+	+
D-galactose	+	+	+	+	+	+	+	+	+	+	+	+	+	+	+	+	+	+	+
D-glucose	+	+	+	+	+	+	+	+	+	+	+	+	+	+	+	+	+	+	+
D-fructose	+	+	+	+	+	+	+	+	+	+	+	+	+	+	+	+	+	+	+
D-monose	+	+	+	+	+	+	+	+	+	+	+	+	+	+	+	+	+	+	+
L-rhamnose	-	-	-	-	-	-	w	-	-	-	-	-	-	-	-	-	-	-	-
D-mannitol	w	w	-	w	w	-	+	-	-	+	+	+	+	+	-	-	-	+	+
N-acetylglucosamine	+	+	+	+	+	+	+	+	+	+	+	+	+	+	+	+	+	+	+
Amygdaline	+	+	+	+	+	+	+	w	w	+	+	+	+	+	+	+	+	+	+
Arbutine	+	+	+	+	+	+	+	+	+	+	+	+	+	+	+	+	+	+	+
Esculine citrate de fer	+	+	+	+	+	+	+	+	+	+	+	+	+	+	+	+	+	+	+
Salicine	+	+	+	+	+	+	+	+	+	+	+	+	+	+	+	+	+	+	+
D-celobiose	+	+	+	+	+	+	+	+	+	+	+	+	+	+	+	+	+	+	+
D-maltose	+	+	+	+	+	+	+	+	+	+	+	+	+	+	+	+	+	+	+
D-lactose(bovine)	+	+	+	+	+	+	+	+	+	+	+	+	+	+	+	+	+	+	+
D-melibiose	+	+	+	+	+	+	+	+	+	+	+	+	+	+	+	+	+	+	+
D-saccharose	+	+	+	+	+	+	+	+	+	+	+	+	+	+	+	+	+	+	+
D-trehalose	+	+	+	+	+	+	+	+	+	+	+	+	+	+	+	+	+	+	+
D-raffinose	w	w	w	w	w	-	w	w	w	+	+	+	+	+	+	+	+	+	+
Amidon	-	-	-	-	-	-	-	w	w	-	-	-	-	-	-	-	-	-	-
Gentiobiose	+	+	+	+	+	+	+	+	+	+	+	+	+	+	+	+	+	+	+
D-tagatose	-	-	-	-	-	-	-	+	+	-	w	-	-	-	-	-	w	-	-
Potassium gluconate	w	w	-	w	-	-	-	-	-	w	w	-	-	-	-	-	w	-	-

+: positive, -: negative, w: weak. Only fermented carbohydrates by at least one strain were mentioned.

**Table 2 foods-12-02312-t002:** Antimicrobial activity expressed as diameter (mm) of inhibition zone exerted by broth cultures (BCs) and treated cell-free supernatants (CFSs) of tested strains (producers) against *K. pneumonia* ATCC 700603, *B. cereus* ATCC 10876, *S. aureus* ATCC 25522, *E. coli* ATCC 43300 and *P. aeruginosa* ATCC 27853 (indicators).

	U21		AB1	49d	ML10	AB5	AB46	4M	AM	FM	QM	MM	ML7	AB36	ML20
*K. pneumonia* ATCC 70060	BC	20.0 ± 0.1	23.00 ± 0.06	19.0 ± 0.06	18.50 ± 0.03	18.50 ± 0.03	15.50 ± 0.03	23.0 ± 0.2	10.0 ± 0.0	12.0 ± 0.2	11.50 ± 0.03	14.6 ± 0.3	23.0 ± 0.1	23.0 ± 0.2	20.0 ± 0.1
	CFS	-	-	-	-	-	-	-	-	-	-	-	-	-	-
*B. cereus* ATCC 10876	BC	16.0 ± 0.1	19.5 ± 0.03	15.0 ± 0.2	22.0 ± 0.1	19.30 ± 0.04	13.5 ± 0.1	20.30 ± 0.04	16.0 ± 0.2	17.50 ± 0.03	14.0 ± 0.1	12.30 ± 0.08	13.0 ± 0.1	17.0 ± 0.3	23.0 ± 0.4
	CFS	-	-	-	-	-	-	-	-	-	-	-	-	-	22.5 ± 0.5
*S. aureus* ATCC 43300	BC	23.0 ± 0.4	20.0 ± 0.1	15.0 ± 0.2	22.3 ± 0.1	17.10 ± 0.05	16.0 ± 0.2	21.0 ± 0.2	14.0 ± 0.4	16.5 ± 0.1	19.0 ± 0.3	19.0 ± 0.2	19.0 ± 0.2	19.0 ± 0.3	18.0 ± 0.1
	CFS	15.0 ± 0.6	-	-	-	-	-	-	-	-	-	-	-	-	-
*E. coli* ATCC 25522	BC	15.0 ± 0.3	12.5 ± 0.1	-	27.6 ± 0.4	14.0 ± 0.2	-	21.0 ± 0.1	-	16.0 ± 0.2	14.6 ± 0.1	14.0 ± 0.06	13.0 ± 0.2	19.0 ± 0.6	18.0 ± 0.1
	CFS	-	-	-	-	-	-	-	-	-	-	-	-	-	10.0 ± 0.2
*P. aeruginosa* ATCC 27853	BC	19.0 ± 0.3	14.0 ± 0.2	-	11.0 ± 0.2	13.3 ± 0.2	-	13.0 ± 0.2	10.6 ± 0.1	14.0 ± 0.2	10.00 ± 0.06	12.6 ± 0.2	08.00 ± 0.06	15.0 ± 0.2	14.0 ± 0.2
	CFS	-	-	-	-	-	-	-	-	-	-	-	-	-	-

-: no activity.

**Table 3 foods-12-02312-t003:** Antibiotic susceptibility profile of indicators expressed as diameter (mm) of inhibition zone.

	*K. pneumonia* ATCC 700603	*B. cereus* ATCC 10876	*S. aureus* ATCC 43300	*E. coli* ATCC 25522	*P. aeruginosa* ATCC 27853
Ampicillin	-	-	-	-	-
Cefoxitin	12.0 ± 0.8	-	10.0 ± 0.1	-	-
Chloranohenicol	12.0 ± 0.3	-	18.0 ± 1.1	-	-
Clindamicyn	-	-	-	-	-
Oxacillin	-	-	-	-	-
Penicillin	-	-	-	-	-
Tetracycline	-	28.0 ± 0.6	23.00 ± 0.06	-	-
Trimethoprim	-	-	16.0 ± 0.5	-	-
Vancomycin	-	-	14.0 ± 1.0	-	-
Erytromicyn	-	-	-	-	-
Naladixic acid	-	-	13.0 ± 0.2	-	14.00 ± 0.08
Streptomycin	19.0 ± 0.9	28.00 ± 0.03	20.00 ± 0.04	20.0 ± 0.3	13.0 ± 0.4
Gentamycin	14.00 ± 0.07	14.0 ± 1.0	23.0 ± 0.3	13.0 ± 0.4	16.0 ± 0.3

-: no activity.

**Table 4 foods-12-02312-t004:** Antibiotic susceptibility profile of tested strains expressed as diameter (mm) of inhibition zone.

	U21	AB1	49d	ML10	AB5	AB46	4M	AM	FM	QM	MM	ML7	AB36	ML20
Ampicillin	S	S	S	S	S	S	S	S	S	S	S	S	S	S
Cefoxitin	S	S	S	I	S	S	I	S	S	S	I	I	I	S
Chloranohenicol	S	S	S	S	S	S	S	S	S	S	S	S	S	S
Clindamicyn	S	S	S	S	S	S	S	S	S	S	S	S	S	S
Oxacillin	R	R	R	R	R	R	R	R	R	R	R	R	R	R
Penicillin	S	R	R	R	R	S	R	S	R	R	I	I	R	I
Tetracycline	R	I	I	R	R	I	I	R	R	R	R	R	I	I
Trimethoprim	S	R	R	R	R	R	I	R	I	I	I	I	I	I
Vancomycin	R	R	R	R	R	R	R	R	R	R	R	R	R	R
Erythromicyn	S	S	S	S	S	S	I	S	S	S	S	S	I	S
Naladixic acid	R	R	R	R	R	R	R	R	R	R	R	R	R	R
Streptomycin	R	I	R	R	I	I	S	S	R	I	R	R	I	S
Gentamycin	R	R	R	R	R	I	R	R	R	R	R	R	R	I

S (sensitive) > 20 mm, 14 mm ≥ I (intermediate) ≤ 20 mm, R (resistant) < 14 mm. (LCLS 2015).

## Data Availability

The data presented in this study are available on request from the corresponding author.

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
