# Peer review of "Characterization of Lactic Acid Bacteria Strains Isolated from Algerian Honeybee and Honey and Exploration of Their Potential Probiotic and Functional Features for Human Use"

_foods, 2023, doi:10.3390/foods12122312_

Round 1
Reviewer 1 Report
The manuscript titled “Exploring some probiotic and functional features of Lactobacillus sp., Apilactobacillus sp. and Fructobacillus sp. strains isolated from Algerian honeybee and honey for human use” has described some potential probiotics strains namely Lactobacillus sp., Apilactobacillus sp. and Fructobacillus sp. strains. The overall work quality of the manuscript seems good. Only some typos error and grammatical error should be resolved. Moreover, the quality of the Fig. 2 should be improved. The title of the manuscript should be revised carefully too.
Author Response
The authors are grateful to the reviewer for the time devoted to read and comment our article.
Point 1: Some typos error and grammatical error should be resolved.
Response 1: done
Point 2: The quality of the Fig. 2 should be improved.
Response 2: done, the quality has been improved.
Point 3: The title of the manuscript should be revised carefully.
Response 3: done, the title has been revised and modified as follow: “Characterization of some lactic acid bacteria strains isolated from Algerien honeybee and honey and exploration of their probiotic and technologic features for human use”.
Reviewer 2 Report
The authors attempted to screen potential probiotics from Algerian honeybee and honey with a relatively large number of evaluation indicators. The experimental method is reasonable and complete, but some parts need further modification and explanation by the authors. I think this paper needs some major revisions.
Detailed comments are as follows:
Some results are described in words only, lacking a very visual graph or table to show them. For example, phylogenetic analysis, hemolysis, biogenic amines production.
Line 16, in vitro needs italics.
Line 21, please add the full name of API 50 CHL.
Line 26, the name of the bacterium needs to be italicized.
Lines 127-128, How can we assess that the isolated strains are capable of producing EPS? Why is sucrose added to the culture medium and is it necessary?
Lines 157-159, what are the selection criteria for the 5 indicator strains in antimicrobial activity?
Line 170, please add a space between for and 15 min.
Line 229, “from Algeria [29,47–49]. .”, please correct it.
Line 302-303, in table 1, the meaning of w is weak, what is the difference between “w” and “W”.
Line 351-352, in figure 1, why the viability of some strains (such as AB70, AB36, ML17, AM, AB24, AB 34) in pH 3 is higher than 0.3% salt solution? Please give an explanation.
Line 374- 375, please add the results of the statistical analysis to the graph.
Author Response
The authors are grateful to the reviewer for the time devoted to read and comment our article
Point 1: Some results are described in words only, lacking a very visual graph or table to show them. For example, phylogenetic analysis, hemolysis, biogenic amines production.
Response 1: we appreciate the suggestion of the reviewer. Phylogenetic analysis was added (L 284). Whereas biogenic amines production and hemolysis, the results are already negative, we think there is no need to rerepresent them graphically or in table.
Point 2: Line 16, in vitro needs italics.
Response 2: thanks for detecting the error. It has now been amended (L17)
Point 3: Line 21, please add the full name of API 50 CHL.
Response 3: done (L22)
Point 4: Line 26, the name of the bacterium needs to be italicized.
Response 4: done, many thanks to detect the mistake (L27)
Point 5: Lines 127-128, How can we assess that the isolated strains are capable of producing EPS?
Response 5: for the first part of the question the sentence “ropy or mucoide large colonies are detected if EPS were produced” (L134).
Why is sucrose added to the culture medium and is it necessary?
Response 5: Several methods are used to screen LAB for EPS producing phenotype:
Detection of “ropy” and “mucoide non-ropy” phenotypes, detection of ropy/mucoid colonies growing in media supplemented with sucrose, rutheniumred staining and viscosity of liquid cultures using “exopolysaccharide selection medium” (ESM).
In this work we were interested in the production not the characterization of the EPS produced (at least at this level). For that, the media with sucrose has been chosen because it gives a clear observation since the colonies are larger when producing EPS, and the phenotype whether ropy or mucoide is much easier detectable.
Point 6: Lines 157-159, what are the selection criteria for the 5 indicator strains in antimicrobial activity?
Response 6: The indicator strains are human pathogens. The sentence “ 5 human pathogenic referenced strains: ” was included (L163).
Point 7: Line 170, please add a space between for and 15 min.
Response 7: done, thanks for detecting the mistake.
Point 8: Line 229, “from Algeria [29,47–49]. .”, please correct it.
Response 8: done, thank you for detecting the mistake.
Point 9: Line 302-303, in table 1, the meaning of w is weak, what is the difference between “w” and “W”.
Response 9: It is a typographical error and it is corrected. Thanks for detecting the error.
Point 10: Line 351-352, in figure 1, why the viability of some strains (such as AB70, AB36, ML17, AM, AB24, AB 34) in pH 3 is higher than 0.3% salt solution? Please give an explanation.
Response 10: Several regulatory mechanisms have been developed by LAB to cope with acidic and biliary stress. Bile salts are synthesized in the liver from cholesterol, cause disruption of cell membranes and lead to DNA damage and oxidative stress (Urdaneta and Casadesús 2017). The presence of a specific enzyme, bile salt hydrolases (BSHs), which cleave bile salts and reduces their toxicity has been described (Foley et al. 2021; Zaghloul and El Halfawy 2022). According to (Rebaza-Cardenas et al (2023), resistance to bile salts seems to be more related to the strains and their origin than to the genus. Since the strains tested in our study are honeybee and heave originated, they are subjected to acidic stress and other type of stressors like high osmolarity generated by high sugar content, and not to bile salt. Otherwise, the tested strains were subjected to simulate gastric conditions for only 2 hours while for 4 hours to intestinal conditions, this may also affected the result of the viability rates.
References
Foley MH, O’Flaherty S, Allen G, et al (2021) Lactobacillus bile salt hydrolase substrate specificity governs bacterial fitness and host colonization. Proc Natl Acad Sci 118:. https://doi.org/10.1073/pnas.2017709118
Rebaza-Cardenas TD, Silva-Cajaleón K, Sabater C, et al (2023) “Masato de Yuca” and “Chicha de Siete Semillas” Two Traditional Vegetable Fermented Beverages from Peru as Source for the Isolation of Potential Probiotic Bacteria. Probiotics Antimicrob Proteins 15:300–311. https://doi.org/10.1007/s12602-021-09836-x
Urdaneta V, Casadesús J (2017) Interactions between bacteria and bile salts in the gastrointestinal and hepatobiliary tracts. Front Med 4:163
Zaghloul HAH, El Halfawy NM (2022) Genomic insights into antibiotic-resistance and virulence genes of Enterococcus faecium strains from the gut of Apis mellifera. Microb Genomics 8:000896. https://doi.org/10.1099/mgen.0.000896
Point 11: Line 374- 375, please add the results of the statistical analysis to the graph.
Response 11: the quality of the graph was improved and the statistical analysis were added also.
Reviewer 3 Report
In the submitted manuscript, the authors have reported the isolation and characterization of fructophilic LAB originated from Algerian honeybee and honey. Although there are some typographical errors in the submitted manuscript, the present study seems to be performed with appropriate methods, thus the manuscript may satisfy the minimal criteria in Foods as scientific paper. However, there are some concerns for publication at present form as follows:
1. The manuscript has showed that many interesting LAB were isolated from the Algerian honeybee and honey, and it seems of interest. However, discussions on following two points seems to be insufficient: a) how each strain plays a role in honeybee or honey based on its specific characteristics; b) what kind of application will be expected by using the isolates with their interesting characteristics. The authors should add substantial discussions and conclusion about those in Results and Discussions and Conclusion sections, respectively.
Minor:
1. Please confirm significant digits. e.g. 20 ± 0.133 -> 20.0 ± 0.1; 12.3 ± 0.08 -> 12.30 ± 0.08
2. Are the antibacterial activity of BCs necessary? Please add reasons/discussion on it.
Author Response
The authors are grateful to the reviewer for the time devoted to read and comment our article.
Point 1: The manuscript has showed that many interesting LAB were isolated from the Algerian honeybee and honey, and it seems of interest. However, discussions on following two points seems to be insufficient:
- How each strain plays a role in honeybee or honey based on its specific characteristics;
Response 1: the information required has been added at lines 44-54.
- What kind of application will be expected by using the isolates with their interesting characteristics. The authors should add substantial discussions and conclusion about those in Results and Discussions and Conclusion sections, respectively.
Response 2: The explanatory sentence has now been included in the manuscript at lines 421-426 and L: 527-534.
Please confirm significant digits. e.g. 20 ± 0.133 -> 20.0 ± 0.1; 12.3 ± 0.08 -> 12.30 ± 0.08
Response 3: done, thank you for detecting the mistake.
Are the antibacterial activity of BCs necessary? Please add reasons/discussion on it.
Response 4: The explanatory sentence has now been included in the manuscript at lines at lines 402-405.
Reviewer 4 Report
Manuscript: Exploring
General comments
The manuscript describes a characterization of some strains belonging four lactic acid bacteria strains isolated from honey of honeybee guts. The origin of the strain is not clear, since the authors mixed both honey and bees gut content. Thus, it is doubtful the real origin of the isolated strains. Some of the trails performed are not clearly described, such as antimicrobial activity. Overall, the manuscript contains a lot of information but disorganized and the findings achieved are not enough to consider a bacterial strain as a “probiotic”. In addition, the manuscript is presented in a disorganized way, with low-quality figures, abundance of spaces mistakes and a clear over abuse of references that makes the work difficult to read and understand. Consequently, I think that the manuscript is not adequate for its publication in Foods.
Specific comments
Line 16: “In vitro” should be written in italics.
Lines 23-25: This conclusion is a very ambitious statement based on the results presented in this article.
Lines 31-33: It seems some confusing idea between the origin of probiotic strain and the way the carry them to consumers. In fact, some probiotic strains were isolated from dairy foods, but the use of dairy foods to carry them is another question different.
Line 34-39. This phrase is very confusing and must be rewritten. Why is necessary to select probiotics by non-dairy foods? The origin is question completely different and unrelated to the way they are administered.
Line 41: LAB should be defined the first time that appears in the text.
Line 90: City of manufacturing and country should be added to “Oxoid”.
Line 116: “Allemagne” is not correct.
Line 124: This is not the most usual method to determine lipolytic activity.
Line 170: Insert and space between “for15”.
Line 171: delete an space previously to “which”. The same in line 309, 310, 320…a lot of different sites of the manuscript.
Lines 156-173. The antimicrobial activity tested are confusing. Where the antimicrobial activity from the LAB strains of metabolites produced by these LABs and isolated? Overnight broth culture of each strain could include also cells and thus, the results obtained is not comparable to standards methods.
Lines 176-190: The method used to determine “cholesterol assimilation” is very low specific. Even assuming that a simple spectrometry can efficiently detect the concentration of cholesterol in a solution, it cannot ensure that the cholesterol has been "assimilated" by bacteria rather than, for example, metabolized or degraded.
Line 229: Delete a dot
Line 24=. Why LAB strains were isolated from honey and bees simultaneously? This seems not make sense, and thus the LAB can not be attributed to bees origin and not to an environmental origin.
Figures 1, 3, and especially Figure 2 are of a very low quality
Author Response
The authors are grateful to the reviewer for the time devoted to read and comment our article.
Point 1: Line 16: “In vitro” should be written in italics.
Response 1: done (L:17), thank you for detecting the mistake.
Point 2: Lines 23-25: This conclusion is a very ambitious statement based on the results presented in this article.
Response 2: In this article we adopted a general procedures for the isolation and characterization of novel strains with presumed probiotic features according to OMS/FAO (2002) recommendations. Accordingly, the isolates must be from natural sources and taxonomically identified at both the genus and species level. After that, the in vitro evaluation is as follow:
1-Characterization of probiotic potential: include resistance to gastric conditions (acid and bile), adherence to mucus and/or human epithelial cells, antimicrobial activity against potentially pathogenic bacteria, modulation of immune system (cytokine profile) and targeted probiotic function.
2-Assessment of safety: include: genome sequencing, antibiotic resistance, assessment of metabolic activities, production of toxic compounds and hemolytic potential.
Testing technological properties is the last level in the in vitro probiotic evaluation, which is not the subject of this paper. There for we consider that the majority of the recommended tests was achieved to presume a potential probiotic features.
Point 3: Lines 31-33: It seems some confusing idea between the origin of probiotic strain and the way the carry them to consumers. In fact, some probiotic strains were isolated from dairy foods, but the use of dairy foods to carry them is another question different.
Response 3: we think that there is no confuse, this idea appeared in the introduction and it is a bibliographic idea. Also it was said that the dairy products are “primarily” “initially” (L: 34) used for incorporation of the probiotics and the common source “commonly” (L: 31)
Point 4: Line 34-39. This phrase is very confusing and must be rewritten. Why is necessary to select probiotics by non-dairy foods? The origin is question completely different and unrelated to the way they are administered.
Response 4: usually autochthonous probiotics are used to formulate foods of the same origins in order to be more effective because it is the same environment, but probiotics from non-dairy products like fermented vegetables or others, could be used in formulation of variety of non-dairy foods.
Point 5: Line 41: LAB should be defined the first time that appears in the text.
Response 5: done (L:42).
Point 6: Line 90: City of manufacturing and country should be added to “Oxoid”.
Response 6: done (L:97).
Point 7: Line 116: “Allemagne” is not correct.
Response 7: done (L: 123). Thank you for detecting the mistake.
Point 8: Line 124: This is not the most usual method to determine lipolytic activity.
Response 8: we used a referenced qualitative method to asses lipolytic activity and we found that it was usually used and the following reference is supporting our choice (Yalçınkaya and Kılıç 2019; Merabti et al. 2019; Saccà and Lodesani 2020)
References
Merabti R, Madec MN, Chuat V, et al (2019) First Insight into the Technological Features of Lactic Acid Bacteria Isolated from Algerian Fermented Wheat Lemzeiet. Curr Microbiol 76:1095–1104. https://doi.org/10.1007/s00284-019-01727-3
Saccà M l., Lodesani M (2020) Isolation of bacterial microbiota associated to honey bees and evaluation of potential biocontrol agents of Varroa destructor. Benef Microbes 11:641–654. https://doi.org/10.3920/BM2019.0164
Yalçınkaya S, Kılıç GB (2019) Isolation, identification and determination of technological properties of the halophilic lactic acid bacteria isolated from table olives. J Food Sci Technol 56:2027–2037. https://doi.org/10.1007/s13197-019-03679-9
Point 9: Line 170: Insert and space between “for15”.
Response 9: done (L: 177)
Point 10: Line 171: delete an space previously to “which”. The same in line 309, 310, 320…a lot of different sites of the manuscript.
Response 10: done
Point 11: Lines 156-173. The antimicrobial activity tested are confusing. Where the antimicrobial activity from the LAB strains of metabolites produced by these LABs and isolated? Overnight broth culture of each strain could include also cells and thus, the results obtained is not comparable to standards methods.
Response 11: the antimicrobial activity was assessed in both BC and treated CFS. An explanatory sentence has now been included in the manuscript at lines at lines 402-405 to determine the metabolites responsible of this activity while the antimicrobial agent in the CFS were discusses in lines 413-447.
Point 12: Lines 176-190: The method used to determine “cholesterol assimilation” is very low specific. Even assuming that a simple spectrometry can efficiently detect the concentration of cholesterol in a solution, it cannot ensure that the cholesterol has been "assimilated" by bacteria rather than, for example, metabolized or degraded.
Response 12: “cholesterol assimilation” was evaluated by a standard referenced method used in almost studies in the field. The reference [44] [21,39] supported this choice.
Line 229: Delete a dot
Response 13: done (235)
Point 14: Line 24=. Why LAB strains were isolated from honey and bees simultaneously? This seems not make sense, and thus the LAB can not be attributed to bees origin and not to an environmental origin.
Response 14: LAB were isolated from the whole intestinal tract or from full honey stomach and from honey separately (L: 91-100). It was to have more diversity in the isolates. Regarding the origin, the honeybee symbionts are found in the whole environment of the heave (beebread, larvae, pollen, flowers….).additionally, all the identified isolates in this paper belong to the 13 honeybee symbionts
Point 15: Figures 1, 3, and especially Figure 2 are of a very low quality
Response 15: the quality has been improved.
Reviewer 5 Report
This is an interesting study that evaluates the probiotic features of various fructophilic LAB isolated from fresh natural honey and honeybees intestinal tract. Since it is important to isolate and characterize LABs from various sources and in many different geographical areas, this study contributes positively to the scientific knowledge in this field.
Some suggestions for improvement:
Line 48: please use full name of A. kunkeei
Table 1: It would be helpful to provide information, (eg a label) about the species that these strains belong and perhaps also about the origin, because the coding eg U21, AB1 is not explained in the text
Line 462: please correct the phrase
Author Response
The authors are grateful to the reviewer for the time devoted to read and comment our article.
Point 4: Line 48: please use full name of A. kunkeei
Response 1: done (L:55), thank you for your suggestion.
Point 2: Table 1: It would be helpful to provide information, (eg a label) about the species that these strains belong and perhaps also about the origin, because the coding eg U21, AB1 is not explained in the text
Response 2: thank you for your suggestion. A label was added (L: 293-296)
Point 3: Line 462: please correct the phrase
Response 3: the information required has been rephrased as follows: “In previous study, Temmerman et al tested the resistance of a multitude of LAB isolates recovered from 55 European probiotic products using the disc diffusion method. The result showed that 79% of the isolates were found to be resistant to kanamycin, 65% to vancomycin, 26% to tetracycline, 23% to penicillin G, 16% to erythromycin and 11% to chloramphenicol [45]”.
Round 2
Reviewer 2 Report
The authors have carefully revised the manuscript and the quality of the manuscript has been improved. Therefore it is recommended to accepted for publication
Author Response
Comments and Suggestions for Authors: The authors have carefully revised the manuscript and the quality of the manuscript has been improved. Therefore it is recommended to accepted for publication
Reviewer 4 Report
During the revision process, the authors have only modified non-substantial aspects that were mentioned in the first round of revisions. They have ratified most of the procedural issues that the reviewer had pointed out to them, such as the lack of consistency in the designation of the strains as probiotics, the fact that cholesterol cannot be considered to have been assimilated instead of degraded or metabolized, or in issues related to sampling or the determination of antimicrobial activity. In most cases, they have ratified the initial version, pointing out that there was some work that had done it in the same way. It is not my role to determine whether such earlier work that the authors cite is well or poorly done, but the expressions the authors use and the conclusions they reach are not supported by the work that has been done. Since the authors ratify and do not amend any of the observations made, I must also ratify and not amend my recommendation to reject the article.
Author Response
Authors thank again the reviewer for reading and correcting this manuscript.
Authors regret that the responses presented in the first round of revision were not sufficient for the reviewer.
Regarding the cholesterol test whether it was assimilated by LABs or degraded, a negative control (uninoculated broth) were used and no change in the initial cholesterol concentration after 24h was detected which confirms its use by LAB strains.
For the rest, all the tests realised in this paper were according to referenced protocols and the consistency in the designation of the strains as probiotics is justified as we targeted only some probiotic traits (the most important).
